



# The European Union Radiological Data Exchange Platform (EURDEP): 25 years of monitoring data exchange

Marco Sangiorgi[1], Miguel Angel Hernández-Ceballos[1], Kevin Jackson[2], Giorgia Cinelli[1], Konstantins Bogucarskis[1], Luca De Felice[1], Andrei Patrascu[1], Marc De Cort[1].

[1]European Commission, Joint Research Centre (JRC), Ispra, Italy
[2]European Commission, Directorate-General for Energy (DG ENER), Luxembourg

*Correspondence to*: Marco SANGIORGI (Marco SANGIORGI@ec.europa.eu)

**Abstract.** During the early phase of an accident with release of radioactive material to the atmosphere having an environmental impact at local or transboundary scale, a rapid and continuous notification and exchange of information including real-time environmental monitoring data to competent authorities and the public is essential to effect appropriate countermeasures. A rapid exchange of information and data must be carried out in a harmonised and consistent manner to facilitate its interpretation and analysis. After the Chernobyl accident in 1986, and in order to avoid that competent
authorities be unprepared again for a similar event, the European Commission defined and put in place a Directive (Council Decision 87/600/EURATOM) which essentially obliges a member state that decides to implement widespread countermeasures to protect its population to notify the European Commission without delay. The same Council Decision also specifies that the results of radiological monitoring must be made available to the Commission and all potentially affected member states. Over the past 30 years, the European Commission has invested resources in developing and
improving a complete system to carry out this delicate task, currently composed of two platforms: the European Community Urgent Radiological Information Exchange, ECURIE, and the European Radiological Data Exchange Platform, EURDEP. This paper aims to increase knowledge of this latter system as a valuable tool to understand and analyse the radioactivity levels in Europe. Commencing with background information, in this paper, we will describe the EURDEP system in detail, with an emphasis on its status, data availability, and how these data are diffused depending on the audience. Within the
scope of this publication, we describe an example of measurements available in the EURDEP system, to be used for scientific purposes. We provide two complete datasets (air concentration samples, http://doi.org/10.2905/23CBC7C4-4FCC-47D5-A286-F8A4EDC8215F (De Cort et al., 2019a) and gamma dose rates, http://doi.org/10.2905/0F9F3E2D-C8D7-4F46-BBE7-EACF3EED1560 (De Cort et al., 2019b)) for the recent radiological release of $^{106}$Ru in Europe which occurred between the end of September and early October 2017. Records stored are publicly accessible through an unrestricted
repository called "COLLECTION" belonging to JRC Data Public Catalogue (https://data.jrc.ec.europa.eu).

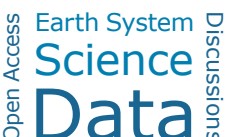

## 1 Introduction

The severe accident occurred at Chernobyl Nuclear Power Plant (NPP) on 26 April 1986 led to a large release of radioactivity to the environment, and large areas of the Europe were contaminated, as can be seen, for example, in the Chernobyl $^{137}$Cs Atlas of Europe (De Cort et al, 1998). Following this accident, much experience was gained in dealing with nuclear emergency preparedness and response, as well as the need to raise awareness of the causes of accidents and to document the lessons that can be learned (IAEA, 2005). This accident and its consequences also led to a large resurgence in radioecological studies aimed at improving remediation and the ability to make predictions on the post-accident situation all in recognition that more knowledge was required to cope effectively with potential future accidents (e.g. (e.g. Alexakhin and Geras'kin, 2011; Beresford et al., 2016).

Among the lessons learned from Chernobyl accident, the large transboundary effects of the radioactive release clearly emphasised the need to establish and support high-level national and international emergency response systems; in particular, countries felt the need to setup activities with a view to significantly improving the exchange of information including monitoring data. The exchange of this information in the early phase of a nuclear or radiological (hereafter, R/N) event, and preferably in real-time, allows competent authorities to be more effective in taking timely and appropriate health and environmental countermeasures to radiation.

Over the past 30 years, the European Commission (hereafter EC) has invested in improving the rapid exchange of information and data during R/N emergencies (De Cort et al., 2011). In 1987, and in order to avoid that the authorities would be as unprepared for future accidents of a similar scale, the EC defined and put in place a Directive (Council Decision 87/600/EURATOM) which essentially obliges a country that decides to implement widespread measures for the protection of its population to notify the EC without delay. The resulting early notification system was named the European Community Urgent Radiological Information Exchange (ECURIE). The same Council Decision also specifies that results of radiological monitoring have to be exchanged. The resulting mechanism for this task was named the EUropean Radiological Data Exchange Platform (EURDEP) (https://remon.jrc.ec.europa.eu/), which was launched in 1994 and is nowadays managed by the EC Joint Research Centre (JRC) sited in Ispra (Italy) as part of their support programme to the Directorate General for Energy (DG ENER). In a platform-wide or European context the EURDEP system does not have a primary alerting role. The notification of a radiological accident or emergency is accomplished through the early notification system ECURIE. Therefore, no actions may be taken based on EURDEP data without prior consultation with the data-provider. The clear concept behind EURDEP is to better equip the decision makers with current and continuous information available in the form of real-time monitoring data to aid in the definition of the most appropriate countermeasures. The EURDEP system is mainly used for the continuous exchange and storage of gamma dose-rate values (hereafter, GDR), and, due to its importance in case of emergency as demonstrated during the Fukushima accident, air concentration levels. Meteorological data, though to a lesser degree, are also exchanged and stored.



Keen interest and high motivation for EURDEP has promoted continuous growth since its creation. Beginning with 6 countries exchanging monitoring data from about 300 stations in various national data formats by e-mail once per week in 1985, EURDEP has evolved to its current state with 39 countries capable of exchanging real-time monitoring information collected from more than 5500 automatic surveillance systems (up to once per hour during an emergency) in a standard data-format through secure ftp and web-services. Such a large-scale harmonised data exchange system for radioactivity measurements is unique in the world. Figure 1 shows the progress of participation in the EURDEP network from 1996 to 2019. It's worth mentioning that the participation in EURDEP for European Union member states is mandatory and the exchange of off-site monitoring data during a radiological accident is an official obligation as required under the 87/600 Council Decision. Participation of non-EU countries in EURDEP is voluntary, but then, in most cases subject to a country-specific Memorandum of Understanding (MoU).

This paper aims to increase knowledge regarding the existence of the EURDEP system as a valuable medium, which can be used for better understanding and analysing of radioactivity levels in Europe. Within the scope of this publication, we aim at describing the unique collection of measurements stored in the EURDEP system since 2002, and at providing an access to relevant datasets to the scientific community. While many of the data have previously been used for various purposes within refereed or scientific applications (e.g. Bossew et al., 2012; 2017; Szegvary et al., 2007), complete datasets for specific radiological events, such as (De Cort et al., 2019a and b) have never been integrally published. In case of a radiological event, this extensive network collects and publishes valuable information to be potentially used for scientific purposes, such as verification of models of source term evaluation and reconstruction, documentation of a temporal and spatial evolution of radioactivity, etc. This information is also useful for people studying risks associated to planned and unplanned releases and for testing models that require spatial data. An example of a dataset within the EURDEP system is the recent $^{106}$Ru release detected in Europe in September-October 2017 (Bossew et al., 2019). In the present paper, we explain how to get access to the full subset of GDR and air samples stored during that period in the EURDEP system.

Section 2 describes the EURDEP monitoring network while Section 3 focuses on EURDEP data, as well as the applied quality control methodology. Section 4 describes the method of exchanging data via the EURDEP system and how data are made accessible to different audiences, while Section 5 presents the data which is available on EURDEP for the 2017 $^{106}$Ru release detected in Europe. Finally, conclusions are reported in Section 6.

## 2. EURDEP networks

The Chernobyl accident resulted in increased radioactivity levels over most of Europe (e.g. Evangeliou et al., 2016). Although several European countries had by 1986 already developed automatic monitoring networks and in some cases had established bilateral agreements for the exchange of this information, the magnitude of the Chernobyl accident (e.g. Steinhauser et al., 2014) demonstrated the need to extend such schemes to the continental scale. Subsequent to the accident, many additional countries set up GDR-based automatic monitoring networks for delineating the radioactive cloud, and in





parallel, decided to, without legal obligations, participate in the international data exchange mechanism EURDEP in order to benefit from the availability of Europe-wide measurements during both routine and emergency.

GDR networks have rather disparate station locations due to differences in the design of national networks. Such differences are in general a consequence of national approaches and policies. The design of the topology of a GDR monitoring network can be based on several factors, such as threat analyses, the enlargement of the area to be monitored, the density of the population and the geological topography of the covered area. In addition, the purpose of the network (alert function only or other functions) and its required technical performance (e.g. spatial resolution) are factors which have an effect on the location of the monitoring sites. As can be seen in Figure 2, countries apply some of the following considerations when siting their monitoring stations:

- Monitoring stations are placed to form a regular grid covering the entire country;
- Monitoring stations are mostly placed at the border of the country, frequently used by countries which do not have national facilities which may cause a radiological release;
- Monitoring stations are mostly placed around sites at risk, i.e. around NPP's;
- Monitoring stations are mostly placed in the surrounding of the most densely populated areas, following the logic that this layout caters for more accurately gauged countermeasures for the local population.

In the early stages of EURDEP development only ADR (ambient dose rate) measurements were exchanged, but over the years other sample types were added. The Fukushima accident showed once more the importance of air concentration data and also – as experienced in Europe - that only HVAS (High Volume Air Samplers) are capable of measuring the extremely low level of contaminants in the air caused by a distant accident. More modern air-concentration sampling stations are fully automated, on-line and connected to the national datacentres so that the data reach EURDEP with the minimum of delay. Most of air sampling stations, however, are off-line and their filters are manually replaced once or twice per week and the measurements are then carried out in laboratories; in this case, the delay with which air-concentration data from these stations are available on EURDEP can be more than a week. To reduce costs and to have nuclide specific data more quickly available, a few European countries have begun using spectrometric probes. Although the sensitivity of this type of probe is much lower than a HVAS station, they can give very valuable results during a local emergency where a higher level of contaminants may occur. As of February 2019, 14 countries exchange nuclide specific data, and among them CH, CY, CZ, DE, EE, FI, HR, NO and LT regularly send air-concentration data which is of utmost importance during an accident.

## 3. EURDEP Data

The existence of the EURDEP platform has contributed significantly to the harmonisation, of the data format and procedures related to the collection of measured radioactivity in Europe in the last two decades. The EURDEP format allows indicating for each measurement whether the data are non-verified (NV), verified (V) or non-plausible (NP). Because the system allows changing this attribute by 'overwriting' already submitted measurements and/or by editing single measurements via the

restricted web-site, the data-providers can send data even before they are verified so as to have the latest measurements available to all users with the minimum possible delay. Hence, most of the measurements presented on the restricted website are non-verified data, which means that meteorological conditions such as heavy rain or snow, or defects in the instruments, electronics or software can result in deviations from the true value. Consequently, isolated seemingly alarming levels on the

map cannot automatically be taken as an indication of increased levels of radioactivity. Even if several nearby stations show such increased values, it does not necessarily imply an actual increase in radiation levels. Some data-providers verify the data and then update the measurements with the "verified" attribute, while other data-providers prefer to leave the data as NV in the system and only to delete or update measurements that did not pass the national verification controls.

### 3.1 Ambient dose rate

ADR measurements (expressed in nSv/h) are crucial to carry out an accurate evaluation of external exposure to human bodies after an accidental or intentional release of radionuclide, as an enhanced external exposure to the population can only partly be assessed from the increase of the measured external net dose rate in a first step (e.g. Sono et al., 2005). Nowadays, EURDEP allows the exchange of the ADR data from more than 5000 on-line monitoring stations in Europe (Figure 2). Most countries send their data within two hours from the measurement, both during routine and emergency, while some countries

send data as a routine on a daily basis, but without any delay during an emergency.

An ideal site for ADR monitoring stations is on extensive flat grassland on natural undisturbed ground, with no obstacles in a circle of at least 20 meters at minimum, and on the height of 1 meter above the ground. In this sense, the way in which a detector is installed strongly influences its readings (e.g. on a wall or a roof in town can give considerably different results from the same probe). In addition, the height above mean sea level of the station, different background levels and probe

characteristics such as the self-effect, contribute to differences in the exchanged measurements that are not relevant to the real radiological situation (e.g. Bossew et al., 2017). In this sense, the availability of the station properties allows a better evaluation of the measurements and estimating the artificial contribution to the ADR, in case of a R/N event, by subtracting self-effect, cosmic radiation and terrestrial background from the reported measurement.

For the purpose of ADR data harmonization is worth mentioning projects such as AIRDOS, MetroERM

(http://earlywarning-emrp.eu/), EGNRS-WG and Intercal (Schauinsland, BfS). The EURDEP system was upgraded in the past years by the so-called AIRDOS extension (Bossew et al., 2007), which provided detailed information on European dose rate monitoring networks focusing on probe characteristics, e.g. the sensitivity and other physical properties. Each station description contains various and sometimes in a very detailed way, characteristics of the probe and the station. This way, corrections in row data can be done at the EURDEP system in a consistent manner using the probe and station properties.

The German BfS developed a simplified site characteristic correction that would allow strongly reducing the measurement deviations caused by the site-specific effects (Stöhlker et al., 2019). It is planned to implement this simplified site characteristic correction in a future EURDEP web-site version so that users can optionally select a view that applies this correction method to obtain a better inter-comparability of the measurements.




### 3.2. Air concentrations

The impact in Europe of the Fukushima accident, although air concentrations remained far below levels which could have caused radiological concern (Bossew et al., 2012), showed the importance of being able to measure the extremely low level of contaminants in the air, caused by either a far-away or small scale accidents. Since 2015, EURDEP stakeholders consider

as a high priority item the exchange of air-concentration data (samples). The recent [106]Ru (Bossew et al., 2019) event has reinforced this need.

Air concentration data which are considered as priority during an accident and are routinely sent to EURDEP are [131]I, [137]Cs, [134]Cs, [132]Te, [7]Be ([7]Be for comparison purpose), [212]Pb and [214]Pb measurements ($Bq/m^3$). These nuclide concentrations are transmitted both during routine and emergency, and the only difference between these two transmissions modality is the

frequency with which such data are sent. Thanks to this improved output, the EURDEP system increases its reliability to track very low levels of radioactivity and contribute in reducing public concern in Europe.

EURDEP also contains data of total-beta radiation ($Bq/m^3$). Currently, there are only 100 stations carrying out this type of measurement. Total-beta measurements are available excluding those related to external-radiation.

### 3.3 Meteorological data

During the years, meteorological data were added to the exchange within the EURDEP system. Weather parameters influence the radiological levels, e.g. atmospheric pressure, snow and rain can change the amount of radon that is released from the soil, and rain can be the cause of higher readings because of radon wash-out effects, while wind strength and direction can help to estimate where and how fast aerosols will move (e.g. Bossew et al., 2017). The scope of this exchange of meteorological information is therefore to facilitate the interpretation of the radiological values, both during routine and

emergency, and to better understand if an increased radioactivity level is caused by natural or artificial events. To this purpose, information about pressure, temperature, wind-direction, wind-speed, precipitation, precipitation-occurrence, precipitation-duration, precipitation-intensity, relative humidity and solar-radiation is exchanged through EURDEP system. However, not all countries deliver this kind of data yet. Countries that send meteorological data (status of February 2019) are the following: BG, BY, CY, CZ, FI, GR, HR, HU, IE, IT, MT, NO, PL, RO, RS and SI. It's worth noting that the

meteorological information transmitted through the network originates from the corresponding EURDEP monitoring station and not from a dedicated one.

### 4 Exchange of EURDEP data and EURDEP maps

#### 4.1. Data exchange

All data exchanged via EURDEP are subjected to copyright of the original data provider and cannot be used for other

purposes, including scientific research, without authorisation.





There are two ways of exchanging radiological data in EURDEP. During routine, the monitoring data is made available by the participating organisations at least once a day, while during an N/R emergency, each organisation makes data available at least once every two hours. In practice, more and more organisations make their national data available on an hourly basis both during routine and during emergency conditions.

An essential condition for exchanging and comparing data at international level is the agreement about some common basic standards about data-format and exchange protocols. All data under EURDEP are exchanged using standard formats and standard exchange protocols which are EURDEP proprietary; since 2013 data format and transmission protocol is compliant with the requirements of the International Radiological Information Exchange (IRIX) format. The IRIX standard, which was developed jointly by the International Atomic Energy Agency (IAEA), the EC and experts from the member states under an

IAEA action plan, is the recommended means to exchange information among emergency response organisations at national and international levels during a nuclear or radiological emergency (Mukhopadhyay et al., 2018). In 2014, in accordance with a 2010 MoU between EC and IAEA which sought to establish a global radiological data exchange system based on EURDEP, EURDEP started the submission of the European radiological data to the International Radiation Monitoring Information System (IRMIS, https://nucleus.iaea.org/Pages/IRMIS.aspx) under provisions given by the Conventions on Early

Notification and Assistance in the case of a nuclear accident or radiological emergency (ENAC). In this way EURDEP assures the role of European Regional HUB for IRMIS. Initially IRMIS was supposed to be constructed on the basis of EURDEP but it was not: IRMIS is currently a data collection tool with a website while EURDEP is a data exchange platform. Also IRIX was modelled on replacing CIS (ECURIE & ENAC) and EURDEP and including other platforms such as INES etc.

Since the exchange of monitoring data at international level plays a fundamental role during an emergency, it is obvious that high availability is a major asset. To address this target, EURDEP has been conceived with high redundancy. The EURDEP network has three central nodes and each central node collects the monitoring data from all the national servers and makes it available on ftp-servers to all participants. The three nodes are located at the EC-JRC in Ispra (Italy), the BfS in Freiburg (Germany) and the EC-DG-ENER (Directorat for Energy) in Luxembourg. The node at the JRC has a special function

because it verifies all the data, loads them in a database and makes them available for viewing and downloading through a web interface. All monitoring data are available and hence can be downloaded from these three sites.

### 4.2 EURDEP maps

As it was commented in the introduction, EURDEP is a tool for decision makers providing notified and continuous real-time monitoring data to define the most appropriate countermeasures before a radioactive plume impacts the population. This fast

and reliable availability of data is important because the radioactive plume can travel over large distances in short time and in any direction, depending on the meteorology during the accident (e.g. the dispersion of radionuclides worldwide from Fukushima Dai-ichi Nuclear Power Plant (e.g. Povinec et al., 2013). To this purpose, the EURDEP Expert Map (restricted website) allows an unlimited access to all monitoring data through various web-services and other secure channels. This map





allows downloading time series, which, in case of an emergency, can be analysed to take the corresponding countermeasures to limit its impact on the population and it has a role in confirming previous predictions and therefore the measures taken.

Nowadays, there is a growing concern with the public about the radioactivity levels, the potential risk of future nuclear accidents and the recovery process in the aftermath of an accident (Sato and Lyamzina 2018). In this context, and based on

the Council Decision 87/600/EURATOM, which also specifies that monitoring data must be made available to the public, a clear and transparent communication is carried out through the EURDEP platform. Most measurements of environmental radioactivity in the form of GDR aggregated averages and maxima for the last 24 hours from some 5500 GDR in 37 European countries are made available through the public freely accessible EURDEP website, which does not require any license nor subscription (Figure 3). This public EURDEP web-site is placed in the Radioactivity Environmental Monitoring

(REM) group website (https://remon.jrc.ec.europa.eu/About). REMon website provides a unique portal of access for all the data and products developed by the REM group at the JRC, such as the European Atlas of Natural Radiation (Cinelli et al., 2019). In addition, and to provide more detailed information to general public and scientists, EURDEP provides to registered users some extra applications, including networks data submission statistics i.e. stations reporting intervals and measurement types, which may help to explain various environmental radioactivity phenomena. These applications, however, may

experiment some malfunctioning due to its transition to a newer version, which is planned to be available in November 2019 and will offer more options in terms of configuration and filtering of environmental radioactivity.

## 5 Data availability for 2017 $^{106}$Ru detection across Europe

Accidents with involvement of radiation sources occur, although infrequently. The IAEA-database of nuclear and radiological incidents can be consulted in http://www.laka.org/docu/ines/, and includes a full list of nuclear and radiation

incidents, which have been reported since 1990 by national nuclear regulatory agencies to the IAEA. In addition, the IAEA database on unusual radiation events (RADEV) is intended, among others, to provide a repository of information on accidents, near-misses and any other unusual events involving radiation sources not directly involved in the production of nuclear power or its fuel cycle.

EURDEP data are collected and stored in routine or emergency mode and constitutes a valuable archive of radiological

measurements potentially used for scientific purposes. We present here an example of two complete datasets referred to a radiological event that can be retrieved from EURDEP database. The example chosen is the widespread detection of $^{106}$Ru in Europe in 2017, which however had not any radiological significance for the population. To point out the event, it is worth mentioning that $^{106}$Ru had only been detected on continental scale following the Chernobyl nuclear power plant accident. Between the end of September and early October 2017, several European networks involved in the monitoring of atmospheric

radioactive contamination, recorded the presence of $^{106}$Ru (Bossew et al., 2019). A possible origin for this $^{106}$Ru inflow to the atmosphere from a ground-based source in the South or Central Urals is substantiated by Shershakov et al., (2019).  Some regions of Central and Eastern Europe measured one-day mean activity concentrations up to 10 over 100 mBq/m³. Different




values for dose conversion factor from activity concentration to air immersion effective dose (EPA 1993; Yoo et al., 2013; TECDOC-1162) would lead to gamma immersion dose rate far below radiological concern and, in addition, below detectability with the EURDEP dose rate network (some nSv/h). For Budapest, as an example, Jakab et al. (2018) estimated an external dose from the radioactive plume (cloudshine) of $9 \times 10^{-8}$ mSv. Figure 4 shows the time series of gamma dose rate, recorded during the event at three measuring sites.

EURDEP data of the [106]Ru event are freely available for download at JRC Data Catalogue, while other datasets of interest may be made available on request. We invite interested people to check regularly JRC data Catalogue (https://data.jrc.ec.europa.eu) for updates and get in contact with the authors for specific requests.

[106]Ru event datasets (total gamma dose rate and sampling stations of air-concentration activity) can be downloaded individually as comma separated values (CSV) files and zipped due to their heavy size by following these subsequent PIDs:

- Ruthenium-106 event (air concentration samples) at http://doi.org/10.2905/23CBC7C4-4FCC-47D5-A286-F8A4EDC8215F (De Cort et al., 2019a)
- Ruthenium-106 event (total gamma dose rates) at http://doi.org/10.2905/0F9F3E2D-C8D7-4F46-BBE7-EACF3EED1560 (De Cort et al., 2019b)

The fields made available for download are the following, in the same order as specified hereafter.

- Gamma dose rates: 2-letter country codes as supplied by the ISO (International Organization for Standardization); monitoring station identifier; locality identifier; longitude; latitude; date and time of begin of measurement (yyyy-mm-dd hh:mm:ss); date and time of end of measurement (yyyy-mm-dd hh:mm:ss); measurement value of gamma dose rate expressed as nSv/h.

- Air-concentration activity: 2-letter country codes as supplied by the ISO (International Organization for Standardization); sampling station identifier; locality identifier; longitude; latitude; sample category identifier radionuclide; measuring unit; date and time of begin of sampling (yyyy-mm-dd hh:mm:ss); date and time of end of sampling (yyyy-mm-dd hh:mm:ss); measurement value of concentration activity.

## 6 Conclusions

The JRC's European radiological data exchange platform (EURDEP) makes radiological monitoring data widely available from most European countries in nearly real-time. EURDEP facilitates the transmission of large datasets from environmental radioactivity and emergency preparedness monitoring networks, as requested by EU legislation between national authorities and the European Commission. It is an integrated part of the official European Commission's radiological/nuclear emergency arrangements which gathers data from 40 networks in 38 European countries plus Canada, mainly through 5500 automatic stations. Even in the current situation there is not a perfect inter-comparability of the ADR data because of the use of different probes, different siting characteristics and monitoring station topologies, but the current degree of harmonisation



and the availability of the station characteristics allow a practical use of the European wide data both in periods with and without events.

Air-sample data communication is still scarce, in comparison with ADR data. Currently, about 14 countries report regularly standard radionuclides ($^{131}$I, $^{137}$Cs, $^{7}$Be etc.). The recent $^{106}$Ru event over Europe, even with a very low radiological impact,
raised the need to improve and increase the exchange of this information in order to deliver a fair picture of the situation and support to decision makers.

EURDEP development activities at the EC are expected to need considerable resources in the future because of a continuous expansion of data-providers and exchanged sample types and nuclides, and it is necessary to keep up with the continuous developments of the large number of national systems which participate to EURDEP. Future work will focus on extending
the number of participating countries, transmitting more on-site meteorological data, expanding to the whole spectrum of sample types foreseen by the Euratom Treaty, refining measurements by applying filters for various natural background components, and, last but not least, globalising the system in collaboration with the IAEA.

**Author contributions**

At the time this paper is written, KB oversees EURDEP development and maintenance and is the main reference for
information being the person responsible of the project; MS and MAHC are the main authors, GC and KJ were the main reviewers and gave technical guidance.

**Competing interests**

The authors declare that they have no conflict of interest.

**Acknowledgments**

We would like to acknowledge the work or the past two decades of the EEWG without which the EURDEP platform would not have developed as it has.

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



**Figure 1. Progress of participating networks in EURDEP. The 40 networks in 2019 are composed of 38 European countries' networks, plus Canada (since 2013) and the DWM-JRC stations located in Ispra, Italy (since 2008). JRC network is operated by EC-staff because it is part of EC territory although located in Italy.**





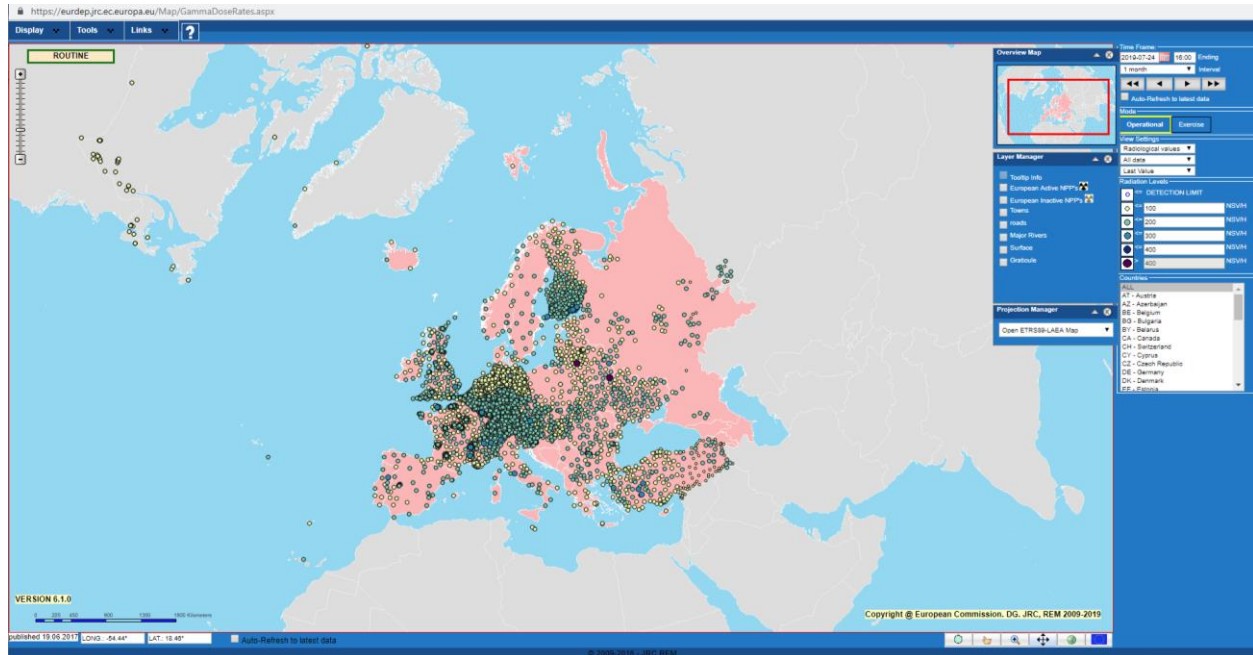

**Figure 2. Screenshot of the EURDEP "expert map" showing the locations of the gamma dose rate monitoring stations. (Copyright @ European Commission. DG. JRC, REM 2009-2019)**



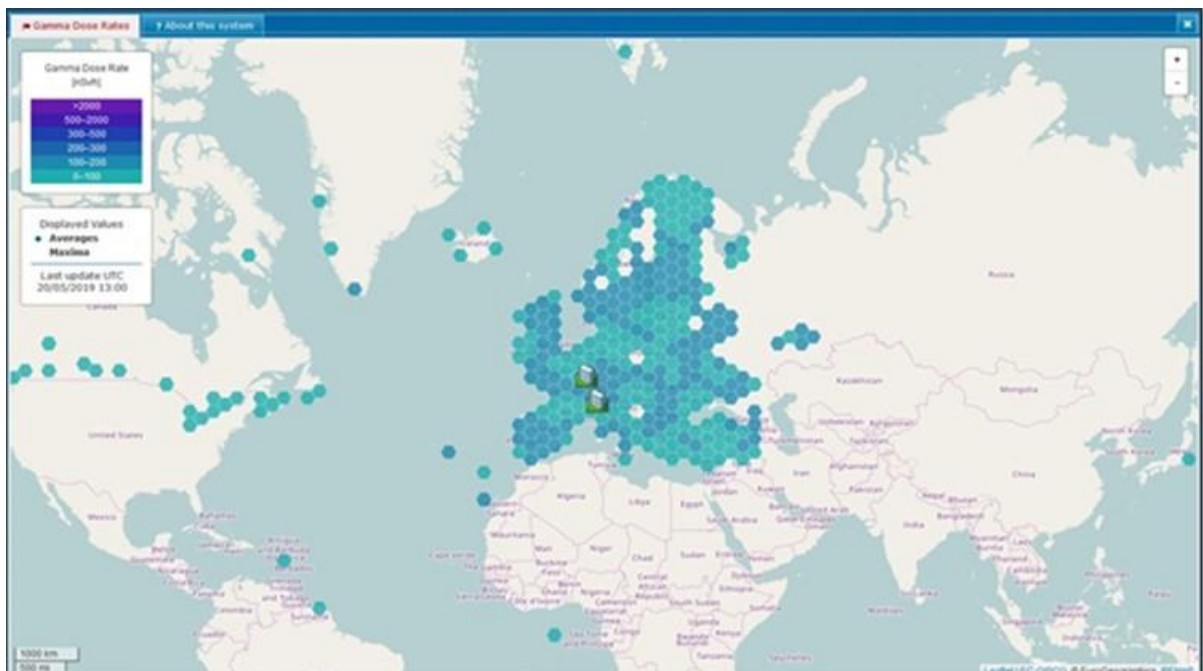

**Figure 3. Screenshot of the EURDEP Public Map. This map shows the maximum and the average value of total gamma dose rate for each hexagon considering the stations inside, which have reported measurements during the last 24 hours and considering values as time-averaged values. (Developed by EC JRC)**

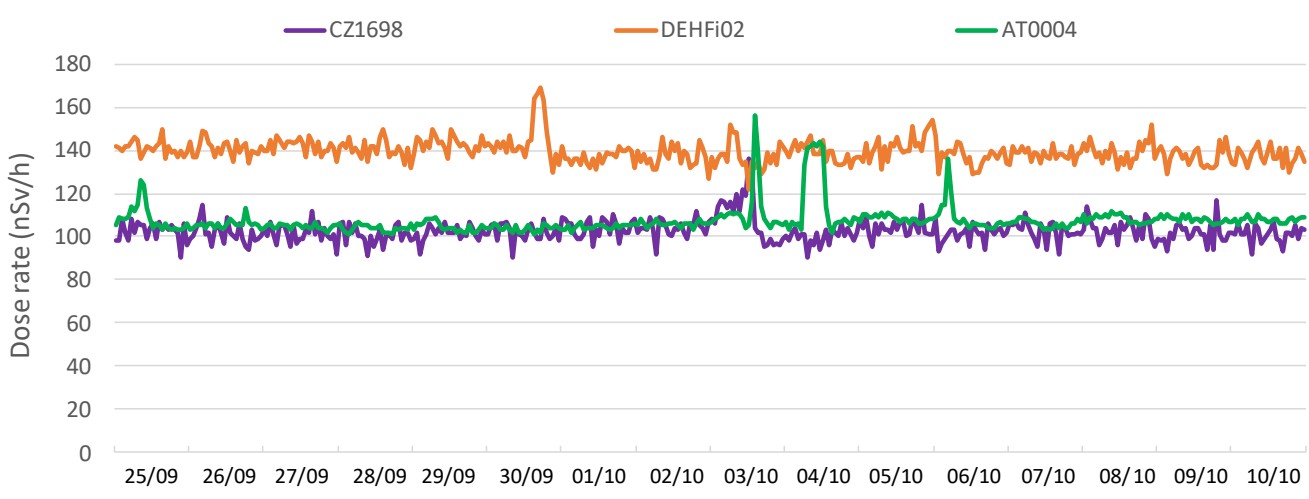

**Figure 4. Time series (day and month of year 2017) of gamma dose rate recorded during the [106]Ru event at three monitoring sites in Europe: one in Czech Republic (CZ1698), one in Germany (DEHFi02) and one in Austria (AT0004).**

