# Peer review of "The European Union Radiological Data Exchange Platform (EURDEP): 25 years of monitoring data exchange"

_Earth System Science Data, 2019_

## Referee Comment (RC1) · Anonymous Referee #1 · 21 Oct 2019

This paper presents the European Union Radiological Data Exchange Platform (EUR-DEP). This data exchange platform is of great interest to general public and scientists. The manuscript is well-written and easy to follow. The EURDEP Public Map is easy to access and the 106Ru data is easy to download and manipulate. The authors also explain why some other data are not publicly available and how to get access to them. There is one minor issue: the EURDEP Public Map only provides the time series of gamma dose rates with lines, but it seems one cannot get access to real numbers at hourly time resolution (the website only provides the daily average, Min and Max GDR at each site). The authors may consider admitting this limitation, and if possible, explaining why and whether the real numbers would be available in the future. I also have

a minor comment on the text on Page 7 Line 18: in this sentence, "IRIX" should be "IRMIS"? and what is CIS and INES?

---

## Referee Comment (RC2) · Anonymous Referee #2 · 29 Oct 2019

This paper discusses The European Union Radiological Data Exchange Platform (EURDEP), which is a important data set for radiological scientists and the public alike. Data is presented effectively, and explanations are given for the degree of publicly available data that exists. Authors also discuss some difficulties concerning national level sampling and the harmonization of that process. Overall, I would accept this paper with minor revisions mentioned in the accompanying PDF.

Please also note the supplement to this comment:
https://www.earth-syst-sci-data-discuss.net/essd-2019-132/essd-2019-132-RC2-supplement.pdf

[Figure]

[Figure]
Overall, an easy to read manuscript with data that is important for early warning systems for radiologic hazards. Just a couple of comments, mostly to improve the reader experience.

Abstract

Line 10: Condense this sentence to improve its readability. It starts the manuscript off in a convoluted manner.

Introduction

Page 2 Line 4-6: Can you explain specifically what experience was gained, or was it more of a realization that we were inadequately prepared, resulting in the need to improve the approach to this type of disaster? Was the experience simply the first time a nuclear disaster had occurred?

Line 31: Some explanation of why the Fukushima accident demonstrates the near for relevant air concentration measuerments would better inform the reader.

EURDEP Networks

Line 18: See above comment about Fukushima

Line 27: Although this manuscript is written for primarily European use, I believe the abbrevations should be defined somewhere, or referred to in a supplement/citation.

Data availability for 2017 106RU detection across Europe

Line 29-30: Any explanation of why this occurred?

**Fig. 1.**

[Figure]

---

## Author Comment (AC1) · 25 Nov 2019

Dear Reviewer,

thank you very much for your very positive comments and the time you have spent in reading this paper, I appreciate it very much.

Regarding to your question about EURDEP Public Map, indeed it was deliberately meant to be as it is and provide less information. Nevertheless, recently we have deployed some improvements which are now online and we invite you to check them. Now two public maps are available: simple and advanced with much more information

available. Nevertheless, the user is not allowed to download data; for that one needs to be registered and granted the permission.

Regarding your comment on the text on Page 7 Line 18, I agree with you it was not clear. I rewrote the whole paragraph:

"Initially IRMIS was supposed to be constructed on the basis of EURDEP but it was not: IRMIS is currently a data collection tool with a website while EURDEP is a data exchange platform. The IRIX protocol was also developed to replace a data-exchange format called CIS (Convention Information Structure) currently used by the European Community Urgent Radiological Information Exchange (ECURIE) platform, and by the prompt notification system called ENAC (Emergency Notification and Assistance Convention) belonging to IAEA. "

---

## Author Comment (AC2) · 26 Nov 2019

Dear Reviewer,

thank you very much for your very positive comments and the time you have spent in reading this paper, we appreciate it very much.

Abstract, Line 10: I rewrote the sentence.

Introduction, page 2, line 4-6: I rewrote the sentence. Nevertheless, it's out of the scope of this paper to explain all lessons learned from Chernobyl accident. We are focused on transboundary effects of the radioactive release which triggered the creation

of ECURIE and EURDEP.

~Line 31: I rewrote the sentence, added a new one and a reference.

EURDEP networks, Page 4, Line 27: I wrote the full countries names

106Ru: I added a reference which may explain why this occurred

———————————————